

# Leaf anatomy and ultrastructure in senescing ancient tree, *Platycladus orientalis* L. (Cupressaceae)

Qianyi Zhou[1], Zhaohong Jiang[2], Xin Zhang[1], Tian Zhang[1], Hailan Zhu[1], Bei Cui[1], Yiming Li[1], Fei Zhao[3] and Zhong Zhao[1]

[1] Key Comprehensive Laboratory of Forestry, College of Forestry, Northwest Agricultural and Forestry University, Yang Ling, Shaanxi, China
[2] College of Life Sciences, Northwest Agricultural and Forestry University, Yang Ling, Shaanxi, China
[3] Beijing Agricultural Technology Extension Station, Beijing, China

Corresponding author
Zhong Zhao,
zhaozhlunwen2010@126.com

## ABSTRACT

*Platycladus orientalis* L. (Cupressaceae) has a lifespan of thousands of years. Ancient trees have very high scientific, economic and cultural values. The senescence of ancient trees is a new research area but is poorly understood. Leaves are the primary and the most sensitive organ of a tree. To understand leaf structural response to tree senescence in ancient trees, experiments investigating the morphology, anatomy and ultrastructure were conducted with one-year leaves of ancient *P. orientalis* (ancient tree >2,000 years) at three different tree senescent levels (healthy, sub-healthy and senescent) at the world's largest planted pure forest in the Mausoleum of Yellow Emperor, Shaanxi Province, China. Observations showed that leaf structure significantly changed with the senescence of trees. The chloroplast, mitochondria, vacuole and cell wall of mesophyll cells were the most significant markers of cellular ultrastructure during tree senescence. Leaf ultrastructure clearly reflected the senescence degree of ancient trees, confirming the visual evaluation from above-ground parts of trees. Understanding the relationships between leaf structure and tree senescence can support decision makers in planning the protection of ancient trees more promptly and effectively by adopting the timely rejuvenation techniques before the whole tree irreversibly recesses.

## INTRODUCTION

Ancient trees are trees that have lived for hundreds or even thousands of years. Compared with other species, trees have always shown exceptional longevity (*Flanary & Kletetschka, 2005*). Ancient trees have very high scientific, economic and cultural values (*Zhu & Lou, 2013*).

The ancient forest which is located in the Mausoleum of Yellow Emperor in Shaanxi Province, China is now the largest, best-preserved planted pure ancient forest of *Platycladus orientalis* L. (Cupressaceae). The world's oldest *P. orientalis* is more than

4,000 years old and is located in the forest (*Chang et al., 2012*; *Zhang et al., 2015*). There are 83,000 *P. orientalis* trees here, more than 30,000 of which are older than 1,000 years. For many years, this forest has been considered a symbol of the long history of the Chinese nation; it is also known as the "soul of the nation" and the "green treasure" composed of "living biological fossils." However, for many reasons, such as climate change and the influence of human activities, the vitality of ancient trees is not optimistic. Many ancient *P. orientalis* trees are sub-healthy or worse. Moreover, there is no simple and effective method for evaluating the senescence degree of these ancient *P. orientalis*.

*Platycladus orientalis* L. (Cupressaceae) is a major tree species for afforestation and reforestation in northern China (*Wang et al., 2016*). It is used as a reforestation species in vulnerable areas because it shows great resistance to severe environmental stresses such as drought, cold, salt and lack of nutrition (*Wang et al., 2005*; *Zhang et al., 2016*). The most impressive characteristic of *P. orientalis* is its extremely high longevity. Many individuals have a lifespan of several thousand years.

The senescence of ancient trees is a new research area but is poorly understood (*Kirkwood, 2005*). The research on plant senescence especially in trees has just started in recent years (*Guarente, 2014*). Although aging and senescence are always mentioned together in animal biology and sometimes used interchangeably, the definition about aging and senescence in plant science are quite different (*Leopold, 1975*). Aging is a key life process along with germination, growth, maturation and death and is usually accompanied by organ recession and decay. Senescence indicates organismal functional decline or physical recession of individuals (*Noodén, 1988*; *Thomas et al., 2003*; *Hongbao, Young & Yan, 2014*). Senescence can occur in individuals at any age (*Hongbao, Young & Yan, 2014*). Therefore, aging is a process, while senescence is a phenomenon affecting individuals.

Trees have an extraordinary long lifespan of more than several thousands of years (*Swetnam & Brown, 1992*). Some extremely old trees maintain a high degree of vitality (*Rajjou & Debeaujon, 2008*; *Thomas, 2013*). For a single tree, the growth curve presents a unique three-stage pattern. At the juvenile stage, the growth rate increases after germination until the mature stage. In this period, trees grow fast to be taller, stronger and mature. This stage will last two to tens of years depending on the tree species. In the mature stage, trees maintain high and strong vitality, constantly producing large numbers of seeds, with a rapid metabolism and a constant renewal of roots and leaves. The growth rate remains relatively stable at a quite high level. This period can last hundreds and even thousands of years. In the senescence stage, trees slow their vigorous growth to a decreased growth rate. This period is very fast compared to the entire lifespan of trees, requiring usually just a few years until complete death (*Issartel & Coiffard, 2011*; *Jones et al., 2014*). Today, research into tree senescence is mainly focused on two aspects: (1) external reasons for long tree lifespan, such as environmental change or environmental stress leading to tree senescence (*Kurepa et al., 2009*; *Zhang et al., 2015*), and (2) internal factors such as genome change or telomere shortening (*Norbury & Hickson, 2001*; *Ally, Ritland & Otto, 2010*; *Song et al., 2011*; *Chang et al., 2017*). Therefore, studies of the longevity traits of trees and their senescence mechanisms are helpful to protect precious ancient trees.

To study the issue of biological senescence, the most critical point is choosing an appropriate and convincing samples (*Becker & Apel, 1993*). Previous senescence-related studies have been greatly influenced by various environmental and genetic factors for each individual. Therefore, to study the senescence-related problems of trees, a combination of tree genetic origins and growing environments can describe the problem more objectively. *Platycladus orientalis* from the Mausoleum of Emperor Huang forest shares similar genetic backgrounds, environments and climate conditions, with an extensive age structure, representing a valuable sampling population for tree senescence-related research.

Leaves are the primary photosynthetic organs in plants and are the part of the plant that directly communicates with the outside world. For annual plants, leaf senescence is accompanied by plant senescence and promotes plant senescence in return (*Dhindsa, Plumb-Dhindsa & Thorpe, 1981*). However, leaf senescence is not equal to whole plant senescence for trees (*Lim, Kim & Gil Nam, 2007*). This phenomenon is especially significant for deciduous trees (*Quy, Zhou & Zhao, 2017*). Unlike deciduous trees, the canopy of evergreen trees (such as *P. orientalis*) is not renewed and disappeared every year, but its leaves still undergo metabolism. Senescence causes a number of different physiological and biochemical changes in the leaves, which often result in changes in cellular structure (*Hengsheng & Yaoqing, 1998*; *Qi & Runze, 2010*). The anatomy of leaves is mainly associated with plant function (*Rossatto & Kolb, 2009*) and always changes with environmental changes (*Fahn, 1986*; *Bosabalidis & Kofidis, 2002*; *Poorter & Bongers, 2006*). Leaf cuticle has a close relationship with plant resistance capability including heat, drought and salinity resistances (*Riederer & Schreiber, 2001*). Chloroplast structure has a significant relationship with photosynthetic function (*Chin & Beevers, 1970*). The intact ultrastructure of chloroplasts guarantees the normal photosynthetic activity of plant (*Kutík et al., 2001*; *Pechová et al., 2003*) because of its protection to chlorophyll, soluble protein, ATP enzyme and PS II center (*Kutík et al., 2000*; *Mitsuya, Takeoka & Miyake, 2000*; *Yamane et al., 2003*).

The leaves of *P. orientalis* are known as scale leaves which are composed of many very small scale-like leaves clinging to small branches in a crossed arrangement (*Hamidipour et al., 2011*). *Platycladus orientalis* tree's leaf branches have no obvious distinction between the abaxial and adaxial which is totally different from typical bifacial leaves such as those of the *Sophora japonica* L. tree (*Quy, Zhou & Zhao, 2017*). *Platycladus orientalis*'s leaf anatomy has a unique symmetrical structure which can be distinguished from other conifer trees (*Hamidipour et al., 2011*). However, certain structural strategies associated with plant restrictions such as anatomy and ultrastructure remain poorly understood (*Coelho et al., 2013*). Researches on plant senescence at the cellular level have been mainly focused on crops so far (*Mitsuya, Takeoka & Miyake, 2000*; *Yamane et al., 2003*; *Vičánková & Kutík, 2005*). Research on leaf structure response to tree senescence is lacking (*Bacic et al., 2004*). A systematic study of the anatomy and ultrastructure of senescence in ancient trees has not yet been reported.

As described above, tree species have a special growth rate curve during their lifespan: the extremely long time of high-level growth even with the increasing tree age, and the

very short time of senescence until the complete death of the whole tree (*Jones et al., 2014*). Thus, we hypothesized that there is an unidentified relationship between tree senescence and leaf structure.

Therefore, we designed this study to identify the relationship among the cellular anatomy, the ultrastructure and the senescence of ancient *P. orientalis* trees. Our objective was to determine the leaf structural response to tree senescence in ancient *P. orientalis*.

# MATERIALS AND METHODS

## Plant material

The ancient forest of *P. orientalis* in the Mausoleum of Yellow Emperor is located on the Loess Plateau, Huangling County, Yan'an City, Shannxi Province, China; latitude 35°34′N, longitude 109°15′E. The annual average temperature is 9.4 °C, and the annual average precipitation is 596.3 mm, while the average annual evaporation capacity is 487.3 mm. This area has between 1,100 and 1,300 annual hours of sunshine, a frost-free period of 170 days, an altitude ranges of 1,100–1,200 m and a typical temperate continental climate with distinct seasonal features. The forest is a planted pure forest of *P. orientalis* with more than 4,000 years of history according to historical records and scientific research. We set up three 50 m by 50 m sample plots. In each plot, we conducted an age investigation and a senescence degree evaluation survey of every single *P. orientalis* tree according to the government records and Table 1.

Sampled ancient *P. orientalis* trees (more than 2,000 years old) were selected from three sample plots. After investigation, we identified three experimental groups which included three senescence degrees (healthy, sub-healthy and senescent). In each experimental group, three sample trees were selected from the three plots (Table 2). As shown in Table 2, detailed information for the nine sampled trees is listed, including basic information, site condition and senescence degree of sampled trees. For each sampling tree, five repeats of the one-year-old scale leaf (new scale leaves grown that year) were selected at a uniform time (between 10 and 12 a.m.) on the sunny side, central crown of the tree without disease or damage.

Leaf samples were collected in July 2014.

## Leaf paraffin sectioning

Leaves were taken from trees at different senescence levels at the same time. After sampling, samples were fixed in formalin-acetic-acid-alcohol fixing solution (5% formaldehyde, 5% acetic acid and 90% of 70% alcohol solution) for three days. They were then dehydrated in graded series of alcohol, infiltrated and embedded in paraffin wax (Sigma 411663; Sigma, Hamburg, Germany) and prepared for paraffin thin sections (*Tosens et al., 2012*). The dehydrating agent used in this experiment was an anhydrous ethanol and tertiary butyl alcohol mixture (*Gu et al., 2014*). The series of transverse thin sections were obtained using a Leica RM2245 semi-automatic rotary microtome. The thickness of all sections were eight μm. Paraffin sections were stained with safranin and fast green dye (*Bryan, 1955*). Samples were observed and images were taken under UOP UB200i microscope (UOP Photoelectric Technology Company, Chongqing, China)

**Table 1 Senescent degree evaluation of trees according to above-ground parts .**

| Assessment items | Evaluation benchmark | | | | | Score |
|---|---|---|---|---|---|---|
| | 0 | 1 | 2 | 3 | 4 | |
| Tree vigor | Vigorous growth | Adversely affected | Apparent weakness | Extremely poor | Almost dead | |
| Tree form | Natural tree form | Nearly natural tree form but some exceptions | Natural tree form partially damaged | Natural tree form damaged and deformed | Natural tree form damaged completely | |
| Branch access | Normal | Have a certain but not obvious influence | Shorter and thinner branches | Branches extremely shortened, internodes inflated | Only have lower growth branches | |
| Upper branches and tree apex mortality | None | Not obvious | Many | A great many | No tree apex and branches | |
| Lower branches mortality | None | Not obvious | Some and some broken | Many, mostly broken | Without healthy branches | |
| Damage of trunk and large branches | None | Rarely and has been restored | Obvious | Obvious and broken | The upper part defect | |
| Foliage density | Branch and leaf density equilibrium | Normal foliage density | Relatively sparse | Few branches, sparse | Dead branches | |
| Size of leaf buds | Leaf (bud) is sufficiently large | Some leaves (bud) smaller | Most buds smaller | All significantly smaller | Only a small number of buds present and smaller | |
| Foliage colors | Almost thick green | Green | Some obvious yellow/brown leaves | Almost light green | All yellow/brown leaves | |
| Bark damage (peeled/necrosis) | No damage | Few holes, no significant damage | Old scars | Wound decayed significantly | Large hole or severe peeling | |
| Bark metabolism | Fresh bark, strong metabolism | Most of the bark fresh, few locations with weak individual metabolism | Apparent lack of vigor, weak metabolism | Almost all bark without vigor | Most of the bark necrotic | |
| Germination and sprouting | Large amount of foliage, much germination and sprouting | Large amount of foliage, some green shoots sprouting | Less foliage, fewer green shoots sprouting | Little foliage, few green shoots sprouting | No germination and sprouting | |
| | Degree of senescence = Total assessment value/Number of assessment items | | | | | Final score |
| Final score | <0.8 | 0.8–1.6 | 1.6–2.4 | 2.4–3.2 | >3.2 | |
| Grade | I | II | III | IV | V | |
| Senescent degree | Healthy | Minor sub-healthy | Sub-healthy | Weak | Senescent | |

**Note:**
Senescent standards of trees.

and measured with Tucsen image analysis system (Tucsen Photonics, Fuzhou, China). A Tucsen image analysis system (Tucsen Photonics, Fuzhou, China) was used to calculate leaf properties such as cuticle thickness, leaf thickness, epidermis thickness, ratios of palisade parenchyma to spongy parenchyma thickness, mesophyll cell thickness and resin cavity width. Fifty corresponding cells per group were measured for anatomical analysis.

**Table 2  Senescent degree and basic information of sampled ancient *P. orientalis* trees.**

| Basic information of sample trees | | | | Site condition of sample trees | | | | Senescent degree evaluation survey judgment of trees | | |
|---|---|---|---|---|---|---|---|---|---|---|
| Government number of ancient tree | Tree age (year) | Height (m) | Diameter at breast height (mm) | Altitude (m) | Slope | GPS coordinates | | Grade | Senescent degree | Sample group division |
| | | | | | | Longitude (E) | Latitude (N) | | | |
| 00225 | >2,000 | 13.6 | 640 | 1111 | 33° | 109°15′42.5″ | 35°35′18.0″ | I | Healthy | Ancient healthy tree |
| 00221 | >2,000 | 15.9 | 607 | 1112 | 10° | 109°15′42.1″ | 35°35′18.1″ | I | Healthy | Ancient healthy tree |
| 00228 | >2,000 | 20.7 | 837 | 1107 | 21° | 109°15′42.9″ | 35°35′17.7″ | I | Healthy | Ancient healthy tree |
| 00360 | >2,000 | 17.8 | 640 | 1104 | 17° | 109°15′42.2″ | 35°35′17.5″ | III | Sub-healthy | Ancient sub-healthy tree |
| 00231 | >2,000 | 12.5 | 572 | 1109 | 26° | 109°15′43.0″ | 35°35′17.5″ | III | Sub-healthy | Ancient sub-healthy tree |
| 00333 | >2,000 | 14.6 | 544 | 1105 | 18° | 109°15′44.1″ | 35°35′18.0″ | III | Sub-healthy | Ancient sub-healthy tree |
| 00227 | >2,000 | 17.6 | 550 | 1094 | 35° | 109°15′43.3″ | 35°35′18.3″ | V | Senescent | Ancient senescent tree |
| 00362 | >2,000 | 11.7 | 630 | 1107 | 15° | 109°15′42.5″ | 35°35′17.2″ | V | Senescent | Ancient senescent tree |
| 00237 | >2,000 | 10.6 | 426 | 1103 | 15° | 109°15′43.7 | 35°35′17.4″ | V | Senescent | Ancient senescent tree |

## Transmission electron microscopy

One-year-old leaves were collected from sample trees and sized at width of natural and length of four mm. Five leaflet replicates from each sampled tree were used for each experimental group. The leaf samples were mixed with a 4% glutaraldehyde fixing solution for preliminary preservation. Then, they were fixed in 2.5% (w/v) glutaraldehyde in 0.1 M phosphate buffer (pH 7.3) overnight (12 h) at 4 °C. Later, after being washed several times with phosphate buffer (pH = 7.3), post-fixation occurred in a 1% solution of osmium tetroxide for 4 h at 4 °C. Dehydration was performed through a graded ethanol series of 30%, 50%, 70%, 80%, 90% and 100% and completed by propylene oxide immersion. The materials were embedded in the Epon 812 resin (Electron Microscopy Sciences, Hatfield, PA, USA). Semi-thin sections of embedded leaf samples were cut on a Leica RM2265 Reichert microtome and initially stained with toluidine blue. Ultrathin sections at 70 nm were cut on a Leica EM UC7 Reichert Ultratome using a diamond knife and post-stained with uranyl acetate and lead citrate (*Kang et al., 1993*; *Ke et al., 2013*). The ultrathin sections on the grid were examined under a Hitachi HT7650 transmission electron microscope (Hitachi Ltd., Tokyo, Japan) operating at 80 kV at different magnifications to obtain the best images. Typically, magnification between 300 and 10,000 was used to observe the ultrastructure of mesophyll cells and cellular organelles such as chloroplasts and mitochondria. Parameters of ultrastructure (cell wall thickness, chloroplast length and width, mitochondria length and width, mitochondria number/cell cross, starch grain length and width, starch grain number/chloroplast) were measured and recorded with 30 independent measurement values for each parameter. Each original data point of cell organelle characteristics was repeated three times.

## Statistical analysis

For anatomical observation and transmission electron microscopy (TEM) observation, statistical analyses were performed using one-way analysis of variance, and the S–N–K test

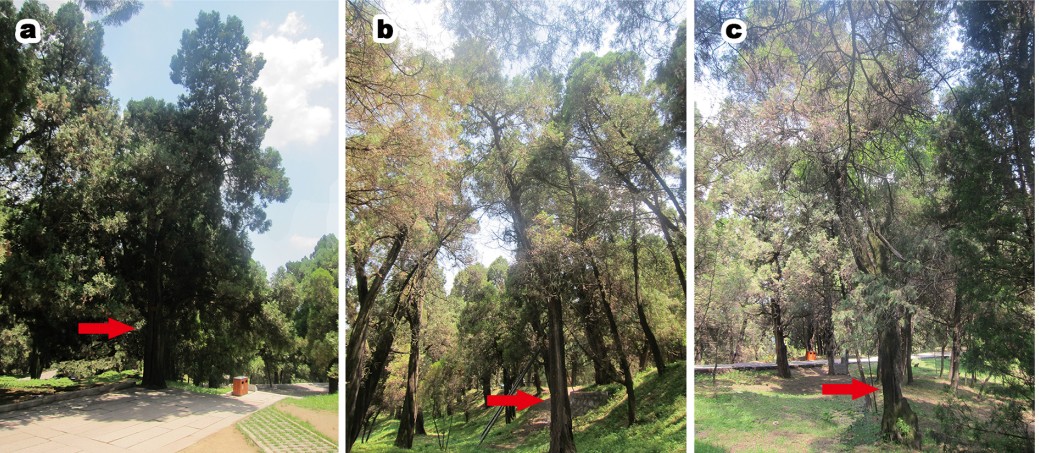

**Figure 1 Ancient *P. orientalis* at different health conditions.** (A) Ancient healthy tree; (B) ancient sub-healthy tree; (C) ancient senescent tree. Red arrows in each images present sampled trees. The photographs in the figure were shot by author Qianyi Zhou.

was used to assess the differences among different experimental groups. The confidence level was set at 95% ($P \leq 0.05$), and the data were displayed as the means ± standard errors (Mean ± SD). The statistical analyses were performed using SPSS 21.0 software (SPSS, Chicago, IL, USA). Microscope images were grouped using Adobe Photoshop CS5 software (Adobe Systems, San Diego, CA, USA) and diagram were drawn using CorelDraw X6 software (Corel Corporation, Dublin, Ireland).

## RESULTS

### Senescence degree evaluations of sampled *P. orientalis* trees

Senescence degree evaluation survey results for three plots showed that the average percentage of ancient *P. orientalis* (only trees older than 2,000 years were included here) at each senescent grade was 13.3% (grade I), 16.7% (grade II), 40.0% (grade III), 23.3% (grade IV) and 6.7% (grade V). More than half of the ancient trees were sub-healthy or worse.

According to the senescence degree evaluation of trees (shown in Table 1), the senescent condition of nine sampled ancient trees and their site situation were measured at the same time (Table 2).

Pictures of *P. orientalis* representing the three experimental groups are shown in Fig. 1. Healthy tree groups (Fig. 1A) were grade I, sub-healthy tree groups (Fig. 1B) were grade III and senescent tree groups (Fig. 1C) were grade V. Seen from the above-ground parts and appearance, healthy *P. orientalis* leaves were green, with complete bark and trunk, with high vitality. Senescent trees have yellow and dark leaf color, hollow and corroded trunks, serious crown loss and serious dieback.

### Leaf anatomy of *P. orientalis* at different tree senescence degrees

Scale leaf anatomical properties of *P. orientalis* at different tree senescence degrees are listed in Table 3. Seven parameters were measured for statistical analysis of anatomy.

**Table 3 Anatomical properties of scale leaves of ancient *P. orientalis* at different tree senescent degrees.**

| Parameters | | Leaf thickness (μm) | Cuticle thickness (μm) | Epidermis thickness (μm) | Spongy parenchyma cell thickness (μm) | Palisade parenchyma cell thickness (μm) | Ratio of palisade / spongy | Resin cavity width (μm) |
|---|---|---|---|---|---|---|---|---|
| Ancient tree | Healthy | $387.11 \pm 46.18^c$ | $5.68 \pm 1.29^a$ | $21.68 \pm 1.78^a$ | $36.53 \pm 2.40^b$ | $61.93 \pm 6.23^b$ | $1.70 \pm 0.21^a$ | $93.60 \pm 11.25^c$ |
| | Sub-healthy | $645.68 \pm 41.79^b$ | $3.91 \pm 0.84^b$ | $18.37 \pm 2.04^b$ | $30.75 \pm 3.10^a$ | $54.01 \pm 5.90^c$ | $1.77 \pm 0.25^a$ | $128.44 \pm 10.09^b$ |
| | Senescent | $1009.85 \pm 19.83^a$ | $3.15 \pm 0.75^c$ | $18.74 \pm 2.82^b$ | $39.00 \pm 3.90^a$ | $68.56 \pm 7.12^a$ | $1.77 \pm 0.24^a$ | $252.93 \pm 29.21^a$ |

Note:
Data are presented as the mean ± SD ($n = 50$). Different lowercase letters under the same parameter denote the least significant differences according to the S–N–K test at $P < 0.05$ compared to data with different senescent degrees but at the same tree age.

The anatomical character of leaf thickness (Table 3) increased with the decrease in tree growth potential. Leaf cuticle thickness (Table 3) of all the ancient trees decreased with the weakened tree growth potential. Compared to all experimental groups, healthy trees exhibited the thickest cuticle. Cuticle thickness declined 2.53 μm from healthy trees to senescent trees. The thickness of epidermis cells (Table 3) decreased from healthy trees to senescent trees in all three groups. Palisade parenchyma cell thickness (Table 3) was much thicker than spongy parenchyma cell thickness in all three experimental groups, but the ratio of palisade/spongy did not show statistically significant differences. Palisade cell thickness had the largest value in senescent trees and the smallest value in sub-healthy trees. The same changes were observed for spongy cell thickness. With the decrease in tree growth potential, the resin cavity (Table 3) increased.

The one-year-old scale leaves were easily recognizable on the *P. orientalis* tree (Fig. 2A). There was no obvious difference in leaf morphology and leaf color among one-year-old scale leaves of ancient *P. orientalis* trees with different senescence degrees just in terms of their appearance (Figs. 2B–2D). However, the senescence degree of trees induced obvious anatomical changes in leaf tissues of *P. orientalis* and significantly reduced the integrity of epidermal and mesophyll cells (Figs. 2E–2F). One-year-old leaves of ancient healthy *P. orientalis* (Fig. 2E) trees had a perfect leaf anatomy with a typical symmetrical structure. Its epidermis were composed of a single layer of epidermal cells. These epidermal cells were small, and the epidermis cuticle thickness was significantly different (Table 3). The central vein of the leaf consisted of vascular cells arranged closely. The resin channels were located on opposite sides of the central vein. The mesophyll cells of scale leaves were composed of spongy tissue and palisade tissue which were spaced closely.

With weak tree growth potential, the anatomy changed dramatically. In the sub-healthy tree group (Fig. 2F), the resin channels were no longer smooth and round in shape. Although the cell shape of epidermal, palisade tissue and spongy tissue cells was normal, there were empty spaces between cells of parenchyma. The destruction and death of protoplasts, and the breakdown of cell membranes were visible.

In senescent tree group (Fig. 2G), the integrity of anatomy was lost and the resin ducts broken into irregular shapes. A large number of cavities appeared between mesophyll tissues, and only a small number of palisade cells was identified; most spongy tissue

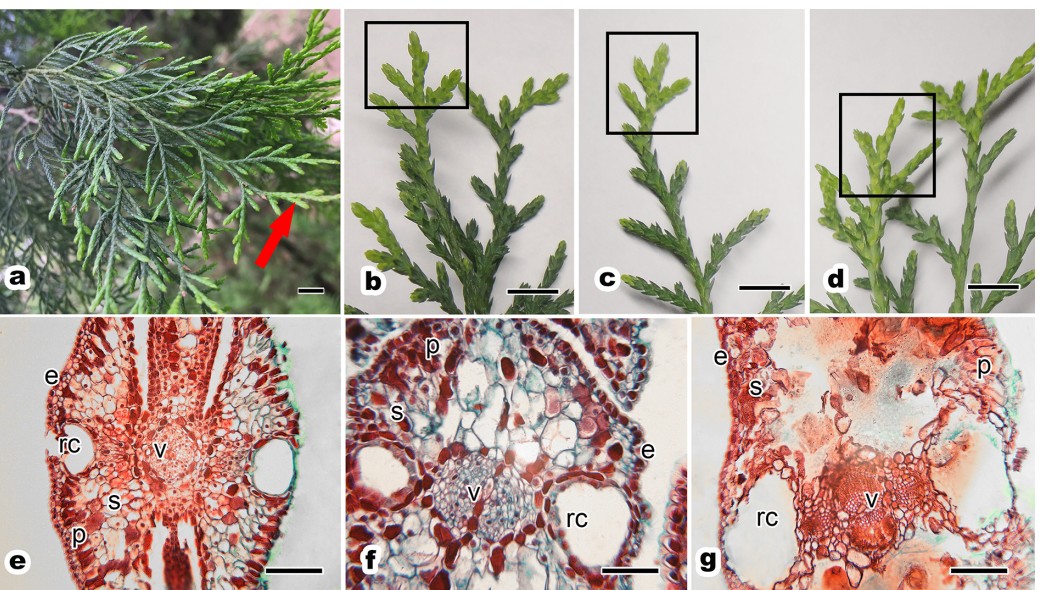

**Figure 2 Morphology and anatomy of one-year-old scale leaf of ancient *P. orientalis* trees with different senescent degrees.** (A) shows a branch and the red shows arrow one-year-old leaves; leaf morphology of (B) ancient healthy tree, (C) ancient sub-healthy tree and (D) ancient senescent tree; one-year-old leaf anatomy of (E) ancient healthy tree, (F) ancient sub-healthy tree and (G) ancient senescent tree. Image on black rectangle in (B–D) corresponds to one-year-old scale leaves. Letters in (E–G) indicating the following: e, epidermis; p, palisade mesophyll cell; s, spongy mesophyll cell; v, vein; rc, resin channel. Scale bars are one cm in (A–D) and are 200 μm in (E–G). The photograph in the figure was shoot by author Qianyi Zhou.

cells had lost their normal structure. The vascular system became disintegrated and fragmented with decreased tree senescence.

## Leaf ultrastructure of *P. orientalis* at different tree senescence degrees

In this research, the size and shape of chloroplasts, mitochondria and starch in chloroplasts of *P. orientalis* at different senescence degrees were observed using TEM. The significant differences in chloroplast, mitochondria and starch are presented in Table 4. Thirty independent mesophyll cells were measured for each parameter.

Chloroplast parameters are presented in Table 4. The length of chloroplasts at different senescence degrees ranged from 2.03 to 4.44 μm, whereas the width of chloroplasts ranged from 3.55 to 4.19 μm. The largest chloroplast length and width were observed in the senescent tree group (4.19 and 2.14 μm, respectively). In the three ancient tree groups, chloroplast length and width showed no significant differences, but the number of chloroplasts per cell cross section in the senescent group was obviously higher than in the healthy and sub-healthy groups. Starch is an accumulation product of photosynthesis in chloroplasts. As shown in Table 4, in the three ancient tree groups, the length of starch grains of healthy trees was slightly higher than in sub-healthy and senescent trees. The number of starch grains declined with senescence degree in ancient trees. Mitochondrial parameters are presented in Table 4. Cross sections of mesophyll cells of *P. orientalis* generally had five to ten mitochondria. The highest number of mitochondria

**Table 4 Ultrastructure properties of mesophyll cells of ancient *P. orientalis* at different senescent degrees.**

| Ancient tree\ Parameters | Cell wall thickness (μm) | Chloroplast length (μm) | Chloroplast width (μm) | Chloroplast number/cell cross section | Mitochondria length (μm) | Mitochondria width (μm) | Mitochondria number/cell cross section | Starch grain length (μm) | Starch grain width (μm) | Starch grain number/ chloroplast |
|---|---|---|---|---|---|---|---|---|---|---|
| Healthy | 0.26 ± 0.03[a] | 3.67 ± 0.61[a] | 2.13 ± 0.71[a] | 5.3 ± 1.25[b] | 0.97 ± 0.60[a] | 0.68 ± 0.32[a] | 10.1 ± 2.08[a] | 2.00 ± 0.62[a] | 1.10 ± 0.53[a] | 1.8 ± 0.42[a] |
| Sub-healthy | 0.21 ± 0.02[b] | 3.55 ± 0.54[a] | 1.70 ±0.32[a] | 4.7 ± 0.48[b] | 0.71 ± 0.15[a] | 0.55 ± 0.12[a] | 8.0 ± 1.15[b] | 1.49 ± 0.29[b] | 0.78 ± 0.23[a] | 1.6 ± 0.70[a] |
| Senescent | 0.19 ± 0.02[b] | 4.19 ± 1.14[a] | 2.14 ± 0.64[a] | 9.7 ± 0.95[a] | 0.67 ± 0.28[a] | 0.52 ± 0.24[a] | 6.0 ± 1.15[c] | 1.54 ± 0.43[b] | 0.79 ± 0.30[a] | 1.0 ± 0.00[b] |

**Note:**
Data are presented as the mean ± SD (*n*=30). Different lowercase letters under the same parameter denote the least significant differences according to the S–N–K test at $P < 0.05$ compared to data with different senescent degrees but at the same tree age.

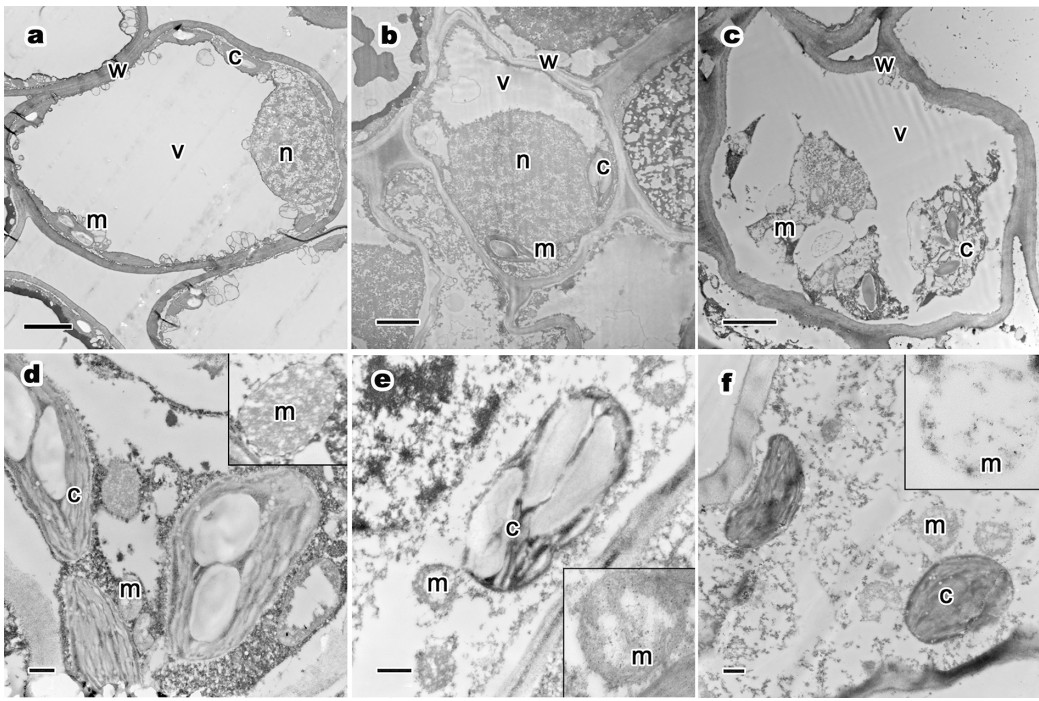

**Figure 3 Ultrastructure of mesophyll cells and organelles, and relative distribution between mitochondria and chloroplasts of *P. orientalis* tree at different senescent states.** (A) and (D), Ancient healthy tree; (B) and (E), ancient sub-healthy tree; (C) and (F), ancient senescent tree. Letters in images represent the following: c, chloroplast; m, mitochondria; w, cell wall; v, vacuole; n, nucleus. Image on black rectangle in picture (D–F) corresponds to enlarge figure of mitochondria of each experimental group. Scale bars in image (A–C) are two μm and in (D–F) are 500 nm.

per cell cross section was 10.1 for ancient healthy trees. The smallest number of mitochondria per cell cross section was 6.0 for ancient senescent trees. The numbers of mitochondria per cell decreased with weakening tree growth condition. For all three experimental tree groups, the length and width of mitochondria showed no significant differences among groups.

The ultrastructure of mesophyll cells and organelles of *P. orientalis* at different tree senescence degrees are shown in Figs. 3A–3C. The main organelles analyzed in this part included cell walls, chloroplast, mitochondria and vacuoles.

Cell wall thickness decreased with weakening tree growth potential in ancient trees (Table 4; Figs. 3A–3C). Healthy trees had the thickest cell wall of all groups (0.26 μm), while the cell wall of senescent ancient trees was the thinnest. Moreover, the degree of bending of the cell wall increased with declining tree health.

The chloroplasts in the mesophyll cells from healthy *P. orientalis* were located at the cell edge, close to the cell plasma membrane and cell wall (Fig. 3A). Most chloroplasts were long rods and bicuspid shaped. In sub-healthy trees, the distribution of chloroplast had changed (Fig. 3B). Some chloroplasts had moved far from the cell plasma membrane, and others were close to the cell wall. However, the distribution of chloroplasts was significantly different in the ancient senescent tree group (Fig. 3C). All chloroplasts in mesophyll cells were distributed near the nuclei. None were close to the cell plasma membrane as in healthy tree groups.

Vacuoles of mesophyll cells in healthy ancient trees were large, centrally distributed and indivisibly, singly monolithic. Sub-healthy trees had many vacuoles of irregular shape distributed near the plasma membrane.

The mitochondria distribution are showed in Fig. 3. The relative distribution of mitochondria and chloroplasts significantly differed among cells from trees in different senescence degrees. In healthy tree groups (Fig. 3D), mitochondria had abundant cristae and were arranged very close to the chloroplasts. In the sub-healthy tree group (Fig. 3E), mitochondria had lost cristae and the distance between mitochondria and chloroplasts was much longer than in healthy trees. In the senescent tree group, mitochondria had lost most cristae and were distributed far from chloroplasts (Fig. 3F). Moreover, mitochondria in senescent tree group did not have intact membranes, as shown in the enlarged images of ultrastructure within black rectangles (Figs. 3C–3E). All mitochondria maintained round or oval shape but with significant changes in cristae. These changes showed that trees' degree of senescence had a great impact on the relative distribution of mitochondria and chloroplasts.

As the most important organelle of plant mesophyll cells, chloroplast apparently changes with tree senescent condition.

*Platycladus orientalis* in the healthy group generally had an intact, smooth, double-layer chloroplast membrane structure (Fig. 4A). The thylakoids had clear, thick stacks and occupied the main interspace of chloroplasts. Low-layered grana lamellae only represented a very small part of the interspace and often appeared as a connection between thick thylakoid stacks. Sub-healthy trees' chloroplast ultrastructure had collapsed synchronously (Fig. 4B). The thylakoids still had a relatively intact membrane system and sharp-edged lamellae structure (images in black rectangles in Fig. 4B). However, chloroplasts had lost their double-layered membrane structure, and the inner contents had leaked out. The number of thick-layered grana stacks had decreased, while thin-layered grana stacks had increased and took up most of the inner chloroplast space. Thus, the thylakoid lamellae were re-arranged from order to disorder when tree growth potential declined from health to sub-health. When tree growth decreased considerably, trees were on the verge of dying and senescence and chloroplasts underwent complete collapse and disintegration (as shown in Fig. 4C). The ultrastructural integrity of both the chloroplast

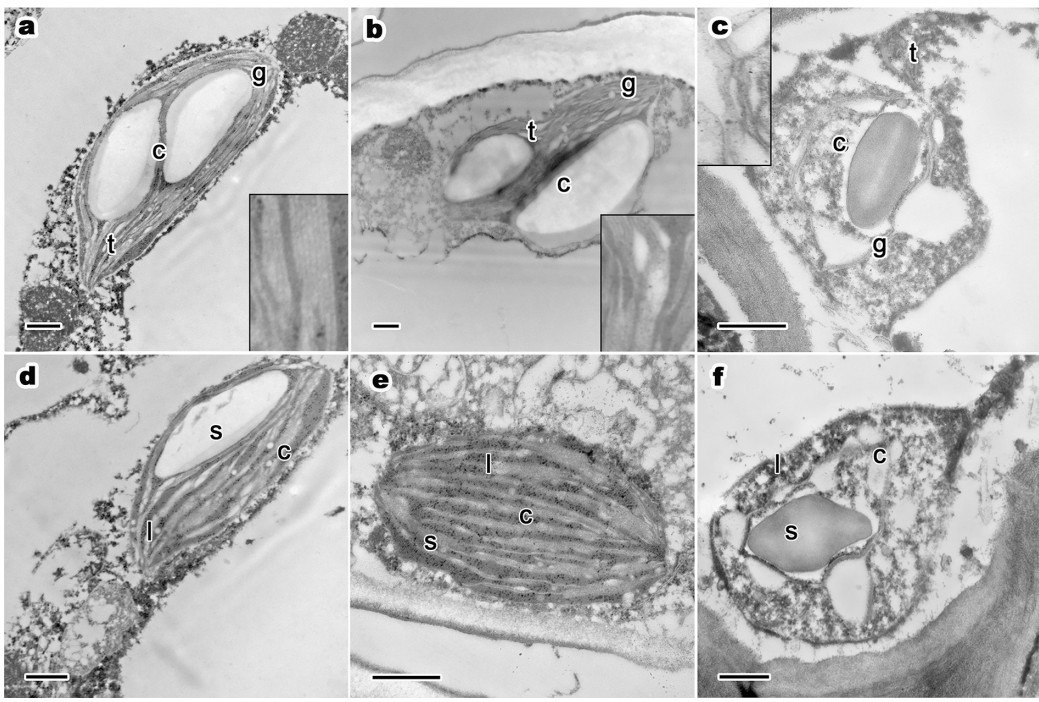

**Figure 4 One-year-old leaf's ultrastructure of chloroplast, lipid droplets and starch grains of *P. orientalis* tree at different tree senescent degrees.** (A) and (D), Ancient healthy tree; (B) and (E), ancient sub-healthy tree; (C) and (F), ancient senescent tree. Letters within images represent the following: c, chloroplast; t, thylakoid; g, grana lamellae; s, starch grain; l, lipid droplet. Image on black rectangle in picture (A–C) corresponds to enlarge figure of chloroplast grana lamellae and thylakoid of each experimental group. Scale bars in all images are 500 nm.

membrane system and the thylakoid membrane was completely lost. Grana presented huge bends and degradation and boundaries between layers, and stacks were fuzzy and illegible. The inclusion of chloroplast had completely diffused to the cytoplasm and the starch granules had remained as large particles.

Osmiophilic granules were the combination of lipid droplets, proesters, ketones and osmium tetroxide (*Stifel et al., 1968*). These stroma-distributed small particles in chloroplasts significantly differed during the tree senescence process. In healthy mesophyll cell chloroplasts of ancient trees (Fig. 4D), lipid droplets sizes were small and evenly distributed in the chloroplast stroma. With weakening tree growth condition, in chloroplasts in sub-healthy trees (Fig. 4E), the distribution of lipid droplets began to concentrate in some regions of the stroma. Moreover, the number of lipid droplets in sub-healthy trees' chloroplasts increased sharply compared with healthy trees. When trees were senescent, the whole chloroplast lost its structural integrity (Figs. 4C and 4F). lipid droplets leaked out to cell cytoplasm with the extremely deformed thylakoid lamellae fractions and chloroplast inclusions.

Most chloroplasts had apparent starch grains. The number of starch grains in chloroplasts changed significantly (as shown in Table 4). Ancient healthy trees had the most starch grains (1.8 starch grains/chloroplast). Most chloroplasts in the other two experimental groups had 1–2 starch grains per chloroplast. The starch grains were

generally huge. In all three groups, starch grains in chloroplasts were distributed in the gap space between grana stacks. There were obvious differences among groups. In ancient healthy tree group, chloroplasts and thylakoid stacks were maintained intact. The presence of starch grains did not affect the normal ultrastructure and integrity of chloroplasts. In the sub-healthy (Fig. 4E) and senescent (Fig. 4F) tree groups, starch grains pressured and even damaged chloroplasts. In summary, starch grains and lipid droplets changed significantly among different tree senescence degrees.

The recession degree of the ancient *P. orientalis* showed consistency with the quality of leaf ultrastructure.

## DISCUSSION

Here, we studied the following main features in one-year-old leaf structure of ancient *P. orientalis* with varying tree senescence degree: (a) anatomical parameter properties and structure of leaf tissue; (b) ultrastructure properties and cellular ultrastructure of mesophyll cells. Chloroplasts, mitochondria and vacuoles were the most vital organelles of mesophyll cells during the senescence of trees according to our research. Combining observations of leaf morphological features, anatomy and ultrastructure of trees under different senescence degrees, there were significant structural differences among trees at different senescence degrees.

Leaf senescence is not equal to whole plant senescence for trees (*Lim, Kim & Gil Nam, 2007*). This phenomenon is especially significant for deciduous trees such as *S. japonica* (*Quy, Zhou & Zhao, 2017*). Unlike deciduous trees, the canopy of evergreen trees (such as *P. orientalis*) is not renewed and disappeared every year, and the leaves still undergo metabolism. Since leaves from previous years cannot reflect the whole tree senescence because of their own structural changes, the well-developed one-year-old leaves with intact anatomy are the best choice to study the leaf structural response to tree senescence (*Gepstein, 2004*).

Leaves are the largest organ of trees that is exposed to the environment. During tree longevity, tree senescence degree is vulnerable to environmental factors such as altitude, temperature, rainfall and soil condition and nutrients (*Wimmer, 2002*; *Peguero-Pina, Sancho-Knapik & Gil-Pelegrín, 2017*). The internal physiology and gene expression are also affected (*Zhang et al., 2015*; *Chang et al., 2017*). These factors can impart regular and obvious morphological and structural changes to the leaves of trees. The structural response can be used to evaluate the overall recession of ancient trees, which can be improved by timely and effective rejuvenation methods.

Tree senescence caused obvious changes in leaf cellular structure. The senescence degree of the tree determined the whole tree's morphology. The senescence of the whole tree is reflected in its leaves (*Lim, Kim & Gil Nam, 2007*). Senescent *P. orientalis* showed a visible decline in appearance including dead branches, a lower density of leaves, yellow foliage, damaged bark and poor growth potential.

The leaf anatomy changed with tree senescence. The significant anatomical structural characteristics of senescent *P. orientalis* included thinner cuticles, decreased epidermal thickness, a wide resin cavity, increased leaf thickness and irregular changes in spongy and

palisade parenchyma cell thickness. The cuticle thickness is closely related to the relative water content of leaves. The thick cuticle helps plants to withstand drought and other abiotic stresses (*Riederer & Schreiber, 2001*; *Yao, Gao & Cheng, 2001*). Moreover, the plant epidermis is a multifunctional tissue playing important roles in relations, defense and pollinator attraction (*Glover, 2000*). The senescent *P. orientalis* had thinner cuticles and thinner epidermal thickness which may indicate poor environmental resistant ability of the whole tree. The leaf resin canal is a mature leaf organizational structure, and the resin acids are helpful for trees to resist diseases, pests and fungi (*Franich, Gadgil & Shain, 1983*; *Rocchini, Lindgren & Bennett, 2010*). However, the too wide and disordered structure of resin channels in senescent trees seriously affected the number and distribution of leaf mesophyll cells (spongy and palisade parenchyma cells), which could lead to the decline of the photosynthetic ability of trees (*Lawson et al., 2003*). These changes fully demonstrate that the physiological function of trees is closely related to their anatomy. Similar results were obtained for anatomical changes of Citrus trees with boron toxicity (*Shao et al., 2014*), for which a significant change was observed in leaf cortex cells and phloem tissue. The spongy parenchyma cells and palisade parenchyma cells were void and distorted (*Shao et al., 2014*). Leaf cuticle is mainly associated with plants' capability to resist environmental stress such as drought, salinity and heat.

Leaves' ultrastructure also changed with tree senescence. All the ancient healthy *P. orientalis* had sharp-edged mitochondria, thickly stacked thylakoids in crescent-shaped intact chloroplasts, a large central vacuole, a large number of chloroplasts, fairly close chloroplast and mitochondria and regular cell walls. In ancient senescent trees, the ultrastructure of mesophyll cells lost its normal structure. The mitochondria cristae were small. The grana stacks were thin and obviously deformed. The chloroplasts were cracked and were far from mitochondria. The vacuoles were broken and decentralized. And cell walls were thin and bent. Chloroplasts are the most sensitive organelle within mesophyll cells. When plants reach senescence, chloroplasts degrade and dismantle before other organelles, leading to a decreased photosynthetic rate (*Hörtensteiner & Feller, 2002*; *Saco, Martin & San Jose, 2013*). In Vlčková's study of the ultrastructure of senescent leaves, the control leaf chloroplast shapes were oval, organized and equally close to the cell wall and the thylakoid layer structure was clear. Senescent leaf chloroplasts were suborbicular and farther away from the edges of the cell wall; the thylakoid layer structure was unclear, with many starch grains accumulated in the chloroplast (*Vlčková et al., 2006*). Normally, starch in senescent tree leaves suggests less effective photosynthesis compared to healthy trees. In our experiment, mitochondria, the other important organelle in mesophyll cells, showed significant structural decline during tree senescence, contradicting to the experiment in which mitochondria remained functional during plant senescence (*Hörtensteiner & Feller, 2002*). Nevertheless, our results agreed with those of Balaban (*Balaban, Nemoto & Finkel, 2005*). Mitochondria have a close relationship with senescence according to the free radical theory (*Shigenaga, Hagen & Ames, 1994*). Reactive oxygen species (ROS) contribute to senescence in both animals and plants (*Trifunovic et al., 2004*; *Guo & Crawford, 2005*; *Noctor, De Paepe & Foyer, 2007*). Reactive oxygen species generated in cells arise in a variety of ways. Among all oxidative burdens

from several cell sources, mitochondria take on the vast majority of cellular ROS burden (estimated at approximately 90%) (*Turrens, 2003*). In senescent plants, mitochondria accumulate large amounts of ROS, which might damage mitochondrial cristae, decrease mitochondrial energy production rate and the metabolism of the mesophyll cell, leaf tissue or the whole plant (*Nemoto et al., 2000*). Most current research on mitochondria ultrastructure and functional changes has been concentrated in algae, crops and model plants such as *Arabidopsis thaliana* (*Romo-Parada et al., 1991*; *Xu et al., 2013*; *Romanova et al., 2016*; *Fanello, Bartoli & Guiamet, 2017*). Woody plants such as *P. orientalis* are completely unstudied in this area. Therefore, the important identified markers of tree senescence in leaf ultrastructure included chloroplast, thylakoid, relative distribution between chloroplast and mitochondria, starch grains, lipid droplets and cell wall according to the observation.

According to the current findings, we built a senescence model of cellular structure change in ancient trees (as shown in Fig. 5). Tree senescence is caused by many factors, which can be divided into internal factors (including gene regulation, species longevity, etc.) and external factors (including environmental stress, human interference, lack of nutrients, damage by disease and insect). Tree senescence changes the cellular structure. The changed cell structure accelerates the rate of senescence of the whole tree in turn. The collapse of the palisade parenchyma cells, spongy cells, epidermal cells, lipid droplets and starch grain accumulation and collapse of chloroplast and mitochondria all accelerates tree senescence. Moreover, cell structure has a close connection with cell function. Chloroplast and mitochondria structure, play a very important role in tree photosynthetic function and growth condition. As a result, when the tree has undergone cellular collapse of the one-year-old leaves, it can only survive for a short time before total death, whereas a healthy tree not only has strong vitality but can also live for a long period even longer than thousands of years. Interestingly, according to our many years of observation, the ancient *P. orientalis* with the significantly changed cell structure were declining and senescing at an alarming rate. These realities were compatible with our research results.

Currently, the majority of ancient *P. orientalis* in Mausoleum of Emperor Huang, Huangling County, Yan'an City, Shaanxi Province, China, are sub-healthy. Very few of the ancient trees are completely healthy, but dying trees also represent only a minority. Thus, timely and effective rejuvenation and protection before most ancient trees in this forest undergo irreversible deterioration is necessary. As these ancient trees represent a precious historical heritage for both scientific research and culture, improving their growth potential and retaining their germplasm resources is urgent.

The relationship between tree senescence and tree leaf structure can support the decision making of when and how to apply rejuvenation techniques such as nutritional supplement, tree pruning, disease control, soil renewal. Moreover, the observation of leaf structure can indicate tree vitality in advance without damage to the precious ancient tree.

Senescence is the final stage of a tree before death and it is determined by complex ecological environmental impacts and internal causes over long periods. Factors leading to tree senescence include structural change, environment stress, physiological function,

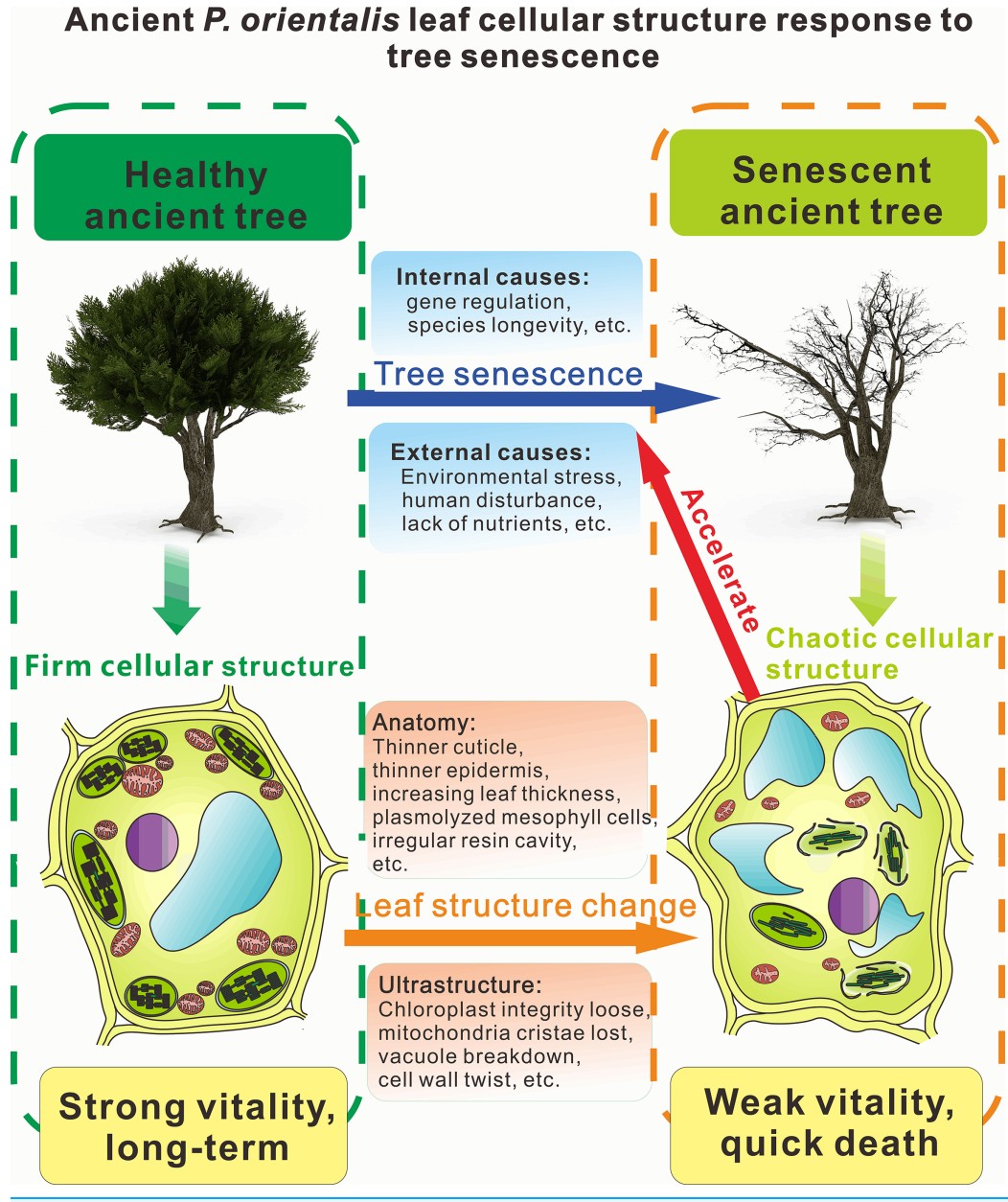

**Figure 5** **Diagram of leaf cellular structural response to tree senescence.** The left part of the diagram is a mesophyll cell structure diagram of an ancient healthy tree, and the right part of the diagram is a mesophyll cell structure diagram of an ancient senescent tree.

biochemistry metabolism and gene regulation. Structure is only the first step of the study, and further research will involve additional experiments (including environmental, physiological, biochemistry, protein and genetic). At present, it is difficult to obtain comprehensive sampling of ancient trees. The accurate age of ancient trees is difficult to determine. Sampling and analysis are difficult because of specificity. Most areas with ancient trees lack relevant data records. For these reasons, current research on tree aging and tree senescence is rare. In future studies, the deep mechanism of structural response to tree senescence will be investigated.

## CONCLUSIONS

The present research studied the leaf structural response of *P. orientalis* (including morphology, anatomy and ultrastructure) to tree senescence. Observations showed that the senescence of trees significantly changed anatomy and ultrastructure of new-grown leaves. The ultrastructure of chloroplast, mitochondria, vacuole and cell wall of mesophyll cells were the most significant markers during tree senescence. The recession degree of the ancient *P. orientalis* showed a consistency with the quality of leaf ultrastructure. Leaf structure study proved the visual evaluation from above-ground parts of trees can be regarded as a quick and efficient way to characterize the senescence degree of trees.

## ACKNOWLEDGEMENTS

It is much appreciated that the Key Laboratory of Agriculture in Arid Areas of Northwest Agricultural and Forestry University (China, Shaanxi, Yangling) and the Key Comprehensive Laboratory of Forestry of Northwest Agricultural and Forestry University (China, Shaanxi, Yangling) provided the experiment platform for us. Authors are grateful to staff of the management region of mausoleum of Emperor Huang and Forestry Bureau of Huangling County.

### Funding

This work was financially supported by the National Forestry Industry Research Special Funds for Public Welfare Projects (China) (201404302). The funders had no role in study design, data collection and analysis, decision to publish, or preparation of the manuscript.

### Grant Disclosure

The following grant information was disclosed by the authors:
National Forestry Industry Research Special Funds for Public Welfare Projects (China): 201404302.

### Competing Interests

The authors declare that they have no competing interests.

### Author Contributions

- Qianyi Zhou conceived and designed the experiments, performed the experiments, analyzed the data, contributed reagents/materials/analysis tools, prepared figures and/or tables, authored or reviewed drafts of the paper, approved the final draft.
- Zhaohong Jiang performed the experiments, analyzed the data, prepared figures and/or tables, approved the final draft.
- Xin Zhang contributed reagents/materials/analysis tools, approved the final draft.
- Tian Zhang performed the experiments, approved the final draft.
- Hailan Zhu contributed reagents/materials/analysis tools, approved the final draft.

- Bei Cui performed the experiments, approved the final draft.
- Yiming Li performed the experiments, approved the final draft.
- Fei Zhao contributed reagents/materials/analysis tools, approved the final draft.
- Zhong Zhao conceived and designed the experiments, authored or reviewed drafts of the paper, approved the final draft.

## Field Study Permissions

The following information was supplied relating to field study approvals (i.e., approving body and any reference numbers):

Field experiments were approved by the management region of mausoleum of Emperor Huang and Forestry Bureau of Huangling County, which is the responsible authority for field studies (#20140607).

## Data Availability

Zhou, Qianyi (2019): Raw_Data. figshare. Dataset. https://doi.org/10.6084/m9.figshare.7575944.v1

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
