# Peer review of "Leaf anatomy and ultrastructure in senescing ancient tree, Platycladus orientalis L. (Cupressaceae)"

_PeerJ, doi:10.7717/peerj.6766_

## Round 0.1 · original submission · Major Revisions

Please address all the comments of both Reviewers very carefully.

Reviewer 1 ·

Basic reporting

The manuscript follows the form of a scientific paper. The English language is reasonably good but needs to be improved. Use of tense is incorrect throughout the manuscript making it difficult for the reader to understand when the authors are describing the results from the current manuscript and results from published papers. There are also minor mistakes or different forms of spelling, or mixing nouns and verbs, which need to be corrected (e.g. ageing – aging, identify – identity, crop – corp).

Introduction describes the context but it is rather long; it can be shortened. For example, it is not necessary to compare human/animal ageing to plant ageing, whereas examples between different type of plants, such as conifers and deciduous trees, are relevant. There is repetition in Introduction, especially when discussing senescence of leaves – part of this could be removed or used in Discussion.

I’d suggest to start the introduction by definition of ancient trees and describing the Mausoleum of Yellow Emperor forest, followed by description of Platycladus orientalis (conifer, evergreen, scale leaves). Then moving into senescense of trees and leaves and the aim of the study.

There are a number of relevant references. It is always the question which references should be left out; one could for example take out the papers referring to human ageing.

Figures and tables are relevant to the study and correctly numbered and referred to in a systematic way throughout the text. However, the image quality is not optimal regarding optimal illumination and resolution (Fig. 2-4).

Experimental design

The manuscript fits in the scope of the journal. The research as such is not very original as senescence of leaves have been studied intensively in different crop plants – but it is justified as a study focusing on the senescence of ancient trees. However, the authors should state more clearly the need of the study. For example, why would you expect that senescence of ancient tree leaves would differ from other plants? Are there emerging problems with vitality of P. orientalis, possibly related to climate change or other environmental factors? The authors could underline even more the importance of the tree species and thus, more detailed knowledge of its anatomy and physiology would gain understanding its life cycle and future management.

Sample collection, preparation and analyses have been performed systematically.

Materials and methods are quite well described, but repetition must be avoided. Some improvement is needed: 1. State more clearly that senescent degree evaluation was performed according to the government standards. 2. State more clearly how the one-year-old leaves were chosen as the right time point is crucial for the study. State also what time of the day the samples were collected – as the amount of starch in chloroplasts depends on the timepoint. 3. I don’t understand the sentence starting on line 154. 4. Line 160-162: State more clearly the composition of FAA. Write ‘graded series of alcohol’ as a mixture of alcohols was used. 5. Is there a reference for staining with Safranin and Fast Green? Otherwise describe the staining. Were they used as counterstains only or as histochemical stains? 6. Line 175: repetition, just write ‘One-year-old leaves were collected…’. 7. Line176: ‘width with 4 mm long’? Line 178-179: ‘0.1 mol l-l phosphate…’? pH 7.3. Line 181: ‘dehydration was performed…’ Line 183: ‘semithin’ rather than ‘half-thin’. Line 189: ‘300 and 10.000’ without mikrometers.

Validity of the findings

The findings are not unexpected – the level of senescence has proceeded from healthy to senescent trees.

It would be good to state more clearly how the one-year-old leaves were identified as the whole study depends on the correct sampling. It would be informative to show the visual morphology of the leaves by adding images of healthy, sub-healthy and senescent shoots to Fig. 2, and pointing out the one-year-old leaves. Does the colour differ between the groups? Also, it would be important to know which time of the day the samples were collected – as you have compared starch granules between the leaf samples.

About light microscopy (Fig. 2): the large opening in the left (Fig. 2b) looks like a resin duct as it is surrounded by parenchymal cells. Are you sure the opening marked with ‘rc’ is a resin duct and not air space between the mesophyll cells? Please, re-check. The same for Fig. 2c – how can you tell which is resin duct and air space? I have not, of course, looked at the samples in microscope where it may be easier to see the difference.

I would prefer changing the order between TEM images, showing Fig. 4 before Fig. 3 as it would help to look at the low-magnification images of the cells before going into details.

In connection of ultrastuctural studies, one should always take to account the possibility (risk) of artefacts due to the extensive sample preparation. For example, the ultrastructure of healthy leaves is not as good as one could expect: cytoplasm is not well preserved and the thylacoid membranes, though clearly having the layered structure, appear as negative (white) towards the background – when they should be dark as both osmium and lead citrate were used in sample preparation. Possibly the initial fixation with a strong 4% glutaraldehyde has blocked the effect of following 2.5% glutaraldehyde and osmium fixation. The internal membranes (cristae) in mitochondria are even less well preserved. Having said this, it is of course important for comparisons that all samples were treated in a similar way.

The comparisons are based on statistics which is fine. However, it may be that it is to some extent overdone. When counting different parametres it is important to remember that you are looking at 2-D sections of cells – you do not have the information for the whole 3-D cells unless keeping track of serial sections. For example, I am hesitant about measuring mitochondria length and width, as they can bend and it is random in which angle and position mitochondria happen to be in a section. I am also hesitant about measuring and counting starch granules as they are also depending on the random sections. Some compensation comes from having a number of cells analyzed.

You should check how you write about the ultrastructure in Results and Discussion – I’d say it is not correct to write about for example ‘number of mitochondria per cell’. Line 289-290: I don’t understand ‘…stacks composed mostly of chloroplast lumen’. Or line 291-292 about electron density and rapid rate of photosynthesis. Line 306-307: osmiophilic granules are not formed during the sample fixation.

In Discussion, there is quite a lot repetition of the results, which could be diminished. Some statements seem opposite to each other such line 389 ‘The senescence of whole trees…’ and line 404 ‘Leaves are a vital organ…’. Discussion has many good elements such as references to the literature about differences between deciduous and conifer trees, or importance of cuticle, but the text needs rewriting. References to the influence of environmental factors on ancient tree anatomy and physiology, and as a cue to protection, might add to the quality of discussion.

Conclusions can be written more clearly. I am hesitant about the last sentence ‘Leaf ultrastucture can be regarded as an efficient way to characterize the degree of senescence of trees.’ The study rather confirmed that the visual evaluation of tree vitality is sufficient. Besides, performing TEM is time-consuming; it can not be an efficient standard method.

Additional comments

It is good to see an anatomical study as understanding changes in anatomy is the key for understanding changes in physiology. The study is performed in a systematic way and the plants studied –ancient trees– are really interesting.

Think about the title: it is not unexpected that the anatomy becomes worse with senescence. I’d suggest a neutral title, for example: Leaf ultrastructure in senescing ancient tree, Platycladus orientalis L. (Cupressaceae). Important to check the abstract and underline the relevance of the study for future management of the ancient tree forest. I would not recommend using the word ‘artificial’ about an ancient planted forest. Fig. 5, a schematic drawing about senescence is usable in oral presentations or in a review article, but I am not convinced it is suitable for the present manuscript. If used, the senescent cell should be clearly plasmolytic.

I’d also recommend to write the whole name of the university in author’s addresses.

·

Basic reporting

The topic of the manuscript is interesting and shows that the authors understand the difficulties of this type of work, and hence have embarked on resolving the problem of senescence degree in the ancient trees with appropriate and many methods, and have tackled the obvious difficulty of tree sampling adequately. However, the manuscript is suffering of poor English language: the tense is wrong in many places in the text and the authors use subjective terms, such as in the title 'leaf anatomical structures become worse'. 'Worse' is not an objective description and hence I would suggest a better formulation of the title in this way 'Advancing senescence is shown in leaf anatomical structure and ultrastructure in ancient Platycladus orientalis L. (Cupressaceae)'. Also, the authors should check the use of subjective terms elsewhere in the text.

Experimental design

Experimental design is of utmost importance in work of this nature. The authors have taken samples of the trees from several experimental plots and several trees in each plot, which seems adequate. In addition, it is stated that the selected branches had no obvious damage by insects or diseases, and the sample scales were taken from mid-crown of the trees, so lower braches were avoided where natural senescence would probably be prevalent.

Validity of the findings

The different degrees of senescence are adequately described, which is vital, and the data seems statistically sound.

Additional comments

The work behind this manuscript is well executed and the data seems sound. I would strongly encourage the authors to have the English language checked by a native speaker.

---

## Round 0.2 · Minor Revisions

Dear Authors,

Reviewer 1 has very carefully gone through your manuscript and submitted a detailed review report. Please revise your manuscript according to the comments of Reviewer 1.

Reviewer 1 ·

Basic reporting

The authors have carefully followed the comments in the previous review. The English language is much improved. However, there is still some fine tuning left to be done.

Instead of ‘morphological structure’ and ‘anatomical structure’ it would be fine to use ‘morphology’ and ‘anatomy’ throughout the manuscript. These terms as such refer to the structure.

The authors have consequently written ‘senescent degree of ancient trees’. I wonder if ‘senescence degree’ would be more correct. Note that ‘senescent tree’ is fine.

Abstract

Line 26-27: I’d suggest ‘…morphology, anatomy and ultrastructure were conducted with one-year leaves of ancient P. orientalis…’

Line 32-33: I’d remove the sentence ‘The recession…’. Then ‘Leaf ultrastructure clearly reflected the senescence degree of ancient trees, confirming the visual evaluation from above-ground parts of trees.’

Line 36-38: Suggestion: ‘Understanding… senescence can support government (decisionmakers?) in planning protection of ancient trees more promptly and effectively, by adopting the timely rejuvenation techniques before the whole tree irreversibly recesses.’

Introduction
Line 69: ‘of’ instead of ‘which’
Line 91: ‘shares’ instead of ‘shared’
Line 113: possibly ‘abaxial and adaxial’ instead of ‘front and back’?
Line 114: ‘from’ instead of ‘with’
Line 115: ‘unique’ instead of ‘unique typical’
Line 119: ‘have been’ instead of ‘are’
Line 122: ‘senescence in ancient trees’

Experimental design

The authors have carefully followed the suggested improvements. Just some details to clarify even better:

Material and methods
Line 141-143: why not to say ‘..conducted an age investigation… of every single P. orientalis tree according to the government standards (Table 1)’.

Line 155: Transfer the meaning ‘Leaf samples were collected in June 2014.’ to the line 151.

Lines 152-154: I don’t understand the meaning. Please, correct.

Line 168: ‘calculate’ instead of ‘calculated’

Lines 173-174: I still don’t understand. Do you mean ‘One-year-old leaves of approximate size 4 x 4 mm were collected…’?

Line 178: ‘was performed’ instead of ‘effected’

Validity of the findings

Again, the authors have carefully taken to account the previous suggestions. Some comments:

Results

Light microscopy

Fig. 2 is much better now, informative and easy to understand. Figure legend have to be improved. For example a) shows a branch and the red arrow one-year-old leaves.

Line 204: I suggest to take off this first meaning.

Line 211: ‘…P. orientalis representing the three experimental groups…’

Line 235: Is it correct to write ‘upper and lower epidermis’ of this type of leaves?

Line 238: ‘vascular cells’ instead of ‘vein cells’

Line 238: You could write ‘The resin channels were located on opposite sides of the central vein.’

Line 242: ‘…smooth and round in shape.’

Line 246: ‘In senescent tree group (Fig. 2g),…’ would be more correct than ‘In senescent tree groups of different tree ages…’.

Line 247: ‘…the resin ducts broken…’

Line 248: ‘…a small number of palisade cells was identified; most spongy tissue cells had lost…’

Line 249: ‘The vascular system became disintegrated and fragmented…’

TEM

The order of TEM images is easier to follow now, good.

Figure 3 legend: ‘nucleus’ instead of ‘nuclear’
Figure 4 legend: reconsider if using ‘osmiophilic granules’ (see below for explanation)

Line 254: not necessary to repeat ‘parameters’

Line 263: ‘Crossections of mesophyll cells…’

Lines: 268-271 Better to write ‘Starch is an accumulation product of photosynthesis in chloroplasts.’ Starch accumulation should be described directly after chloroplast ultrastucture. Mitochondria after that.

Lines 273-274: ‘…cell walls, chloroplasts, mitochondria and vacuoles.’

Lines 281: ‘…bicuspid shaped. … distribution of chloroplasts had changed.’

Line 282: ‘…chloroplasts had moved…’.

Line 287-288: Maybe easier to write ‘Sub-healthy trees had many vacuoles of irregular shape distributed near the plasma membrane.’

Line 289: The sentence ‘Situation…’ can be removed or modified. Do not use ‘worse’.

Line 293: ‘Mitochondria had lost…’

Lines 294-295: Remove ‘Mitochondrial membranes showed unclear conditions…’

Line 296: ‘mitochondria had lost…’

Line 299: ‘oval shape’

Lines 301-302: Modify. ‘Chloroplast ultrastructure’ is not the organelle.

Line 307: ‘..had collapsed’

Line 309-311: ‘…chloroplasts had lost… contents had leaked out… stacks had decreased … had increased…’

Line 311: ‘Thus, the thylakoid lamellae were re-arrenged from order to disorder..’

Line 317-318: Modify the sentence ‘Chloroplast inclusion…’

Line 319-320: You would need a reference to the sentence ‘Osmiophilic granules… osmium tetroxide’. Though at the current study, the osmiophilic granules do not show as black as they should when treated with osmium. Here the granules are shown as white against the background, which is fine as they can be seen. Maybe the authors should consider using some other word instead of ‘osmiophilic granules’, such as ‘lipid droplets’.

Line 322: ‘..sizes..’

Line 329: ‘Strach grains in chloroplasts changed significantly…??’

Line 339: avoid using ‘amazing’

Discussion

Line 343: ‘studied’ instead of ‘detected’

Line 353-354: ‘renewed and disappeared.. and the leaves still… Since leaves from previous years can not…’

Line 357: ‘Leaves are the largest organ…’

Line 369: ‘The leaf anatomy…’

Line 371: ‘..increased leaf thickness..’ instead of ‘thick leaf thickness’

Note: when you discuss about chloroplast structure and photosynthesis, you could mention that less starch in senescent tree leaves suggest less effective photosynthesis compared to healthy trees.

Line 373: together with Yao et al. 2001 you could already refer to Riederer & Schreiber 2001 here, instead of later

Lines 387-390: I don’t understand, modify or remove the two sentences: ‘The ratio…shape changes.’

Line 394-397: Modify (shorten) the sentence ‘In ancient senescent trees…’

Line 397: ‘Chloroplasts are…’

Line 400: ‘In a study…’ you have to refer directly to Vlckova et al. 2006

Lines 406-408: Modify, better not to write Hörtensteiner & Feller twice. What do you mean by ‘Toren and Rober’? Is it a reference? In case, why do you mention Balaban et al. 2005?

Line 418: Arabidopsis thaliana in italics

Line 420: ‘senescence’ instead of ‘senescent’

Line 431: ‘longer’ instead of ‘older’

Line 432: you should consider some other word than ‘worst’

Lines 435-443 are repeating things written before; try to combine and shorten the text at p. 11

Line 450-451: You could remove the sentence ‘The recession degree…’

Line 452: Maybe you could state something like ‘The relationship… structure can support the decision making of when and how to apply rejuvenation techniques…’

Line 455: remove ‘etc.’

Line 457: I’m not sure why you say ‘Senescence is the final external performance…’

Line 460: ‘future research’ or ‘further research’ instead of ‘additional research’

Lines 462-464: I’m not sure if you need to take up details in denied sampling.

Lines 464-465: I don’t understand the meaning of ‘The accurate age… internationally.’

Lines 465-466: I don’t understand the meaning of ‘Some areas… lack relevant data records.’ In China? Or in general – would not it be ‘many areas’ or ‘most areas’?

Compare lines 468-474 and Conclusions (lines 477-482): you could combine and shorten these two parts into Conclusions.

Check Figure 5. Minor changes are needed:
‘Firm cellular structure’ instead of ‘Good..’?
‘Anatomy’ instead of ‘anatomical structure’
‘thinner epidermis’ instead of ‘…epidermal thickness’
‘increasing leaf thickness’ instead of ‘thick…’
‘plasmolyzed mesophyll cells’ instead of ‘irregular…’
‘irregular resin ducts’ instead of ‘wide & irregular resin cavities’

Ultrastructure
‘loose’ or ‘lost’ instead of ‘lose’ and ‘lack’
‘breakdown’ or ‘breakage’ instead of ‘break’

‘quick death’ instead of ‘quick to death’

Additional comments

The suggestions above are of minor characteristics to improve the manuscript even more. They should not take long for the authors to check. I trust the authors have double-checked that all references used in the text, and only them, are included in the reference list.

---

## Round 0.3 · accepted · Accept

Dear Authors,

You have carefully and successfully addressed the remaining comments of Reviewer 1.